# Will Human-Induced Vegetation Regreening Continually Decrease Runoff in the Loess Plateau of China?

**Yanzhong Li [1]** **, Dehua Mao [2],\*** **, Aiqing Feng [3] and Tayler Schillerberg [4]**

[1] School of Hydrology and Water Resources, Nanjing University of Information Science and Technology, Nanjing 210044, China; liyz_nuist@163.com

[2] Key Laboratory of Wetland Ecology and Environment, Northeast Institute of Geography and Agroecology, Chinese Academy of Sciences, Changchun 130102, China

[3] National Climate Center & Laboratory for Climate Studies, China Meteorological Administration, Beijing 100081, China; aiqingfeng2011@163.com

[4] Department of Crop, Soil and Environmental Sciences, Auburn University, AL 36849, USA; tas0053@tigermail.auburn.edu

\* Correspondence: maodehua@iga.ac.cn

**Abstract:** China has become the largest contributing country to global vegetation regreening. However, the regreening pattern and subsequent impact on arid areas have not been comprehensively evaluated. Therefore, we selected the Loess Plateau, a representative arid region that has undergone evident vegetation restoration, to investigate the spatial patterns and temporal trends, as well as the drivers of vegetation change. This study primarily focused on 12 afforested watersheds during 2000–2018. Furthermore, both the impacts of vegetation regreening on runoff for the past two decades and the future projections were quantified based on the fraction of photosynthetically active radiation ($f$PAR), the Budyko model, and the global climate models (GCMs). $f$PAR for the last two decades indicates that vegetation in the Loess Plateau has experienced a continuous increasing trend during the growing season, primarily in response to the implementation of the Grain for Green Project (GFGP). Of the 12 watersheds, 9 experienced significant $f$PAR change with a change rate above 50%, and 11 exhibited a significant increase ($p < 0.05$) in runoff sensitivity to vegetation regreening, which indicates that vegetation regreening plays an increasingly important role in controlling runoff variation. The decline in runoff caused by vegetation regreening was particularly noticeable before 2011 or 2012; afterwards, runoff tended to vary with precipitation. In the future (2020–2049 and 2050–2099), decrease in runoff by regreening will be limited, as runoff is anticipated to decrease by 3.5% in 2020–2049 and 4.1% in 2050–2099 with a 20% increase in $f$PAR. These results indicate that runoff tends to be stable even with continuous vegetation regreening. While the reduction of runoff by regreening will be limited in the future, rapid human-induced vegetation regreening may aggravate water scarcity when flash droughts occur and may result in disasters in water-limited regions to the socio-economic stability and agriculture. Our study will provide an applicable theoretical foundation for water resources decision-making and ecological restoration.

**Keywords:** remote vegetation index; Loess Plateau; runoff change; regreening; Budyko model; GCMs

## 1. Introduction

The Loess Plateau is located in northern China's semi-arid and semi-humid climate transition zone. The region is home to the world's most severe soil erosion and water loss occurrence, with an average annual soil loss of 5000–10,000 t km$^{-2}$. Wind-deposited loess soils, low vegetation coverage,

high-intensity summer rainstorms, and a long history of intense agricultural activities are responsible for the Loess Plateau's current condition [1,2]. In an attempt to address this ecological issue, over the past decade, the central government of China has adopted numerous soil conservation strategies including terraces, check dams, reservoirs, and reforestation [3,4]. The most famous one is the Grain for Green Project (GFGP), which has been implemented since 1999 and is the largest regreening or reforestation project worldwide [5]. Despite the effectiveness of the GFGP's measures in controlling soil erosion and ecological degradation, their mitigation methods also sharply reduced the observed runoff and induced a freshwater deficit in this eco-fragile area [1,6]. Therefore, it is necessary to assess the vegetation regreening modification and its impact on runoff over the Loess Plateau.

Vegetation regreening has been tracked and mapped throughout the world at multiple scales using remote sensing, due to the advantages of high temporal resolution and broad spatial coverage [7,8]. The TERRA satellite was launched at the end of 1999, outfitted with the moderate resolution imaging spectroradiometer (MODIS) and multiangle imaging spectroradiometer (MISR) instruments. Thus, it provides a more consistent and moderate spatial (500 m) and temporal (8 days) resolution for vegetation products, via the fraction of absorbed photosynthetically active radiation ($f$PAR) obtained by MODIS. The $f$PAR was derived from a synergistic algorithm used in MODIS-only and MISR-only models [9]. Since then, the $f$PAR vegetation parameter that characterizes the vegetation canopy functioning and energy absorption capacity has been widely used in ecosystem productivity models, global climate models, and hydrological models [10].

Vegetation, which is a key component of land cover, plays a crucial role in the hydrologic cycle [11]. Changes in vegetation, such as regreening or reforestation, can alter water balance factors or modify how precipitation partitions into runoff by affecting canopy interception, leaf evaporation, infiltration, and soil-water storage, thereby influencing the hydrological processes and water availability [12]. Thus, quantifying the influence of vegetation change on runoff is critically important to water resource management and designing vegetation restoration strategies. In general, there are three approaches to quantifying the impacts of vegetation change on runoff: (1) paired watersheds experiment, (2) hydrological model, and (3) runoff sensitivity analysis [13]. The runoff sensitivity analysis, coupled with the Budyko hypothesis and performed with fewer parameters and simple mathematical expressions [14,15], has been considered an effective method for quantifying the influences of vegetation change on runoff [16–18]. Li and Pan [19] assessed the impact of vegetation change on runoff in 26 large watersheds (>50,000 km$^2$) around the world by constructing an explicit relationship between parameter $w$, representing the water use efficiency of vegetation, and Normalized Difference Vegetation Index (NDVI). However, such a relationship does not work well for small-scale watersheds. Recently, Zhang and Yang [10] established an explicit relationship between the $f$PAR change and landscape parameter $n$ in the Budyko equations at various watershed scales. $f$PAR change can be directly converted into a change of parameter $n$, and thus provides a good method for quantitatively assessing the hydrological response to vegetation regreening in the Loess Plateau.

In this study, we selected 12 mostly afforested watersheds in the Loess Plateau of China. The objectives are to (1) detect the spatiotemporal regreening change characterized by $f$PAR during 2000–2018, as well as the underlying regreening driver; (2) investigate the runoff's sensitivity variations to $f$PAR change and the relative contribution of regreening to runoff changes in the past two decades; and (3) quantify the relative contribution of $f$PAR to runoff changes in future decades (2020–2099) under five scenarios of vegetation regreening, based on global climate models (GCMs). The findings in this study are expected to support regional water resource management and improve ecological restoration.

## 2. Material and Methods

### 2.1. Study Area

The Loess Plateau is located in northern China (Figure 1) and covers an area of $640 \times 10^3$ km$^2$, or 6.7% of China's land. The plateau is characterized by a semi-humid and semi-arid climate,

with an aridity index (AI) ranging from 0.9 in the southeast regions to 11 in the northwest regions (Figure 2a). The soil type is dominated by entisol with an average depth in excess of 100 mm (Figure 2b). Silt clay-loam texture is common, with a more sandy texture in the northwest and more clay in the southeast (Figure 2c). Complex geomorphic landforms are located in the southwest, south, and middle regions of the Yellow River (Figure 2d). Examples include, but are not limited to plateaus, ridges, mounds, and gullies with steep slopes. In contrast, a relatively flat area is located in the northwest region with a slope of less than 5° (Figure 2d).

The most severely eroded areas of the Loess Plateau are located along the middle Yellow River [20], i.e., from the Hekou to Longmen (Figure 1), and thus were considered critical areas for the GFGP. As such, 12 mostly afforested watersheds in the middle Yellow River were selected to investigate the influence of regreening on hydrologic responses. The spatial distribution of the 12 selected watersheds is shown in Figure 1, and the hydro-meteorological characteristics are listed in Table 1.

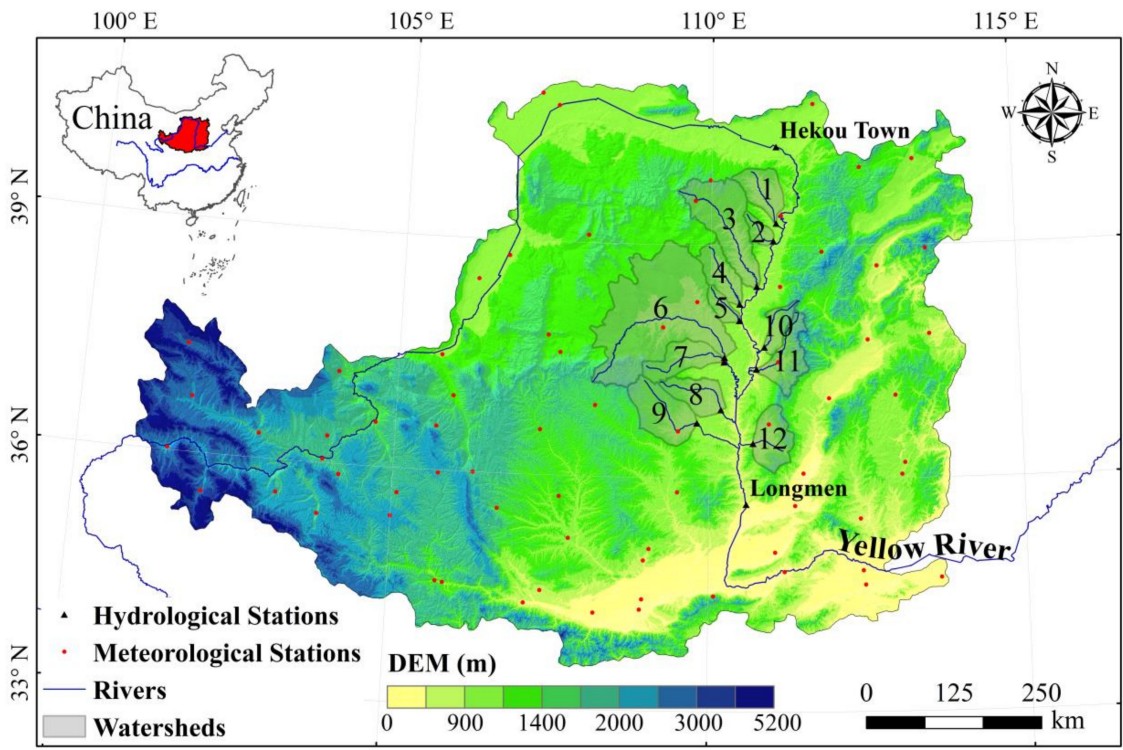

**Figure 1.** Location of the Loess Plateau and the 12 selected watersheds.

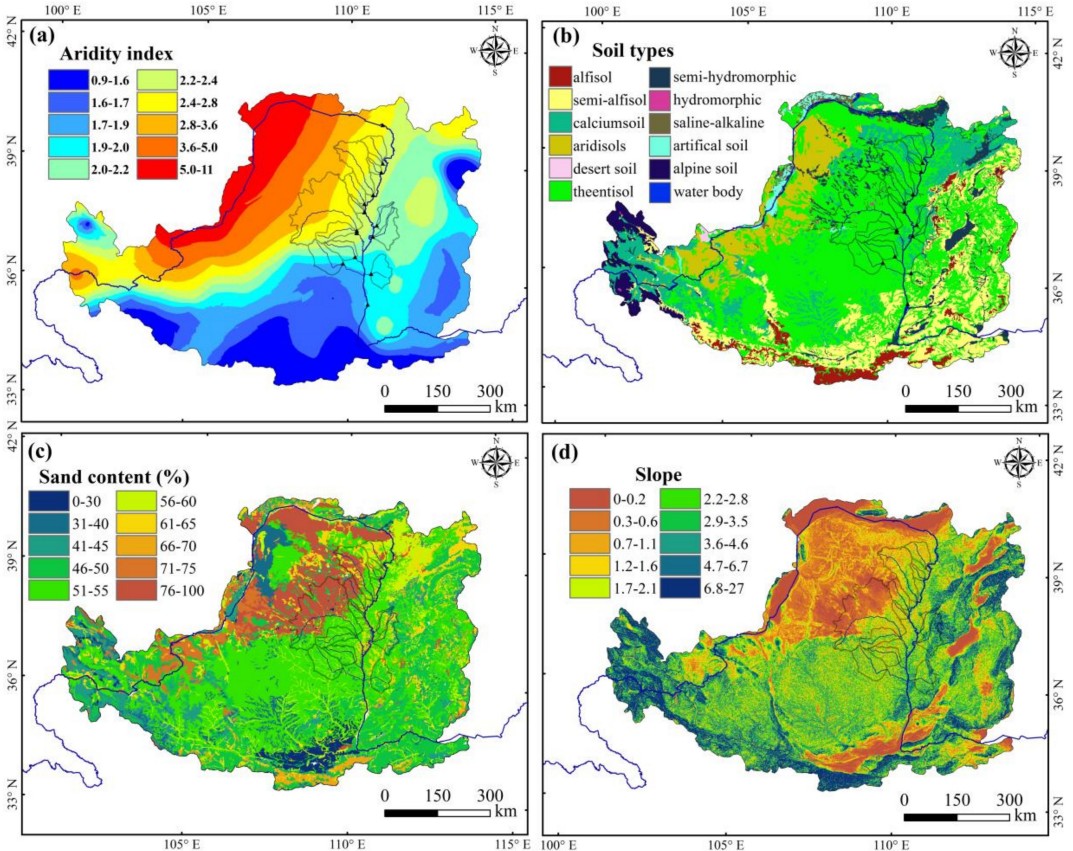

**Figure 2.** Climate, soil, and terrain characteristics: (**a**) aridity index, (**b**) soil types, (**c**) sand content, and (**d**) slope.

**Table 1.** Basic hydro-meteorological characteristics of the 12 selected watersheds in the Loess Plateau.

| Watershed ID | River Name | Hydrological Station | Area (km$^2$) | $R_{obs}$ (mm) | $P_{re}$ (mm) | PET (mm) | AI |
|---|---|---|---|---|---|---|---|
| 1 | Huangfu | Huangfu | 3175 | 11.22 | 402 | 981 | 2.44 |
| 2 | Gushan | Gaoshiya | 1263 | 14.29 | 428 | 988 | 2.31 |
| 3 | Kuye | Wenjiachuan | 8515 | 25.37 | 409 | 1021 | 2.49 |
| 4 | Tuwei | Gaojiachuan | 3253 | 65.79 | 436 | 1024 | 2.35 |
| 5 | Jialu | Shenjiawa | 1121 | 29.11 | 451 | 1024 | 2.27 |
| 6 | Wuding | Dingjiagou | 23422 | 27.45 | 411 | 1032 | 2.51 |
| 7 | Dali | Suide | 3893 | 27.13 | 457 | 1010 | 2.21 |
| 8 | Qingjian | Yanchuan | 3468 | 23.66 | 488 | 987 | 2.02 |
| 9 | Yanshui | Ganguyi | 5891 | 25.95 | 488 | 977 | 2.00 |
| 10 | Qiushui | Linjiaping | 1873 | 20.09 | 474 | 990 | 2.09 |
| 11 | Sanchuan | Houdacheng | 4102 | 36.32 | 484 | 977 | 2.02 |
| 12 | Xinshui | Daning | 3992 | 21.18 | 512 | 961 | 1.87 |

Notes: $R_{obs}$ refers to the annual runoff depth, $P_{re}$ refers to the annual precipitation, and PET refers to the potential evapotranspiration calculated by the Penman-Monteith method. AI refers to the aridity index which is defined as the ratio of PET to $P_{re}$. The time range is 2000–2017.

## 2.2. Data Sources

### 2.2.1. Runoff and Climate Data

Monthly runoff records at the hydrological stations were obtained from the Hydrological Bureau of the Ministry of Water Resources of China. Daily meteorological records for the 72 national meteorological stations from 2000 to 2017, including precipitation, temperature, relative humidity, wind speed, sunshine duration, and vapor pressure, were acquired from the National Climatic Center of China Meteorological Administrator (http://data.cma.cn/). Biases in precipitation have been widely recognized due to station distribution, observational techniques, and gauge measurement error,

and such biases can introduce significant uncertainties in the data application. Thus, a bias correction to precipitation was implemented by using the following equation:

$$P_c = \frac{\left(P_g + \Delta P_W\right)}{CR} \qquad (1)$$

where $P_c$ is the bias-corrected precipitation, $P_g$ is the daily precipitation from the gauge observation, and $\Delta P_w$ is the wetting loss. *CR* is the catch ratio defined as the ratio of gauge precipitation to the true precipitation. A larger *CR* value indicates that more precipitation is caught by the instrument. Figure 3a shows the remarkable difference in CR spatial distribution, with values spanning from 0.82 to 0.98. The corrected precipitation was calculated according to the *CR* values over the Loess Plateau (Figure 3b). The daily precipitation was then aggregated to annual values, interpolated into the space grid, and extracted by the 12 watersheds polygon to get the watershed precipitation.

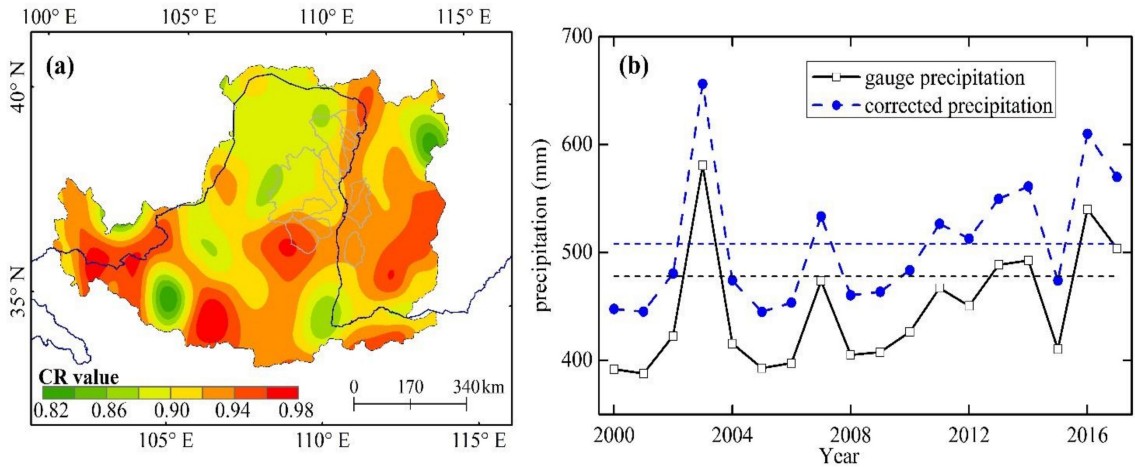

**Figure 3.** Bias correction to the precipitation over the Loess Plateau: (**a**) catch ratio distribution; (**b**) the changes in gauged and bias-corrected precipitation.

Four typical global climate models (GCMs) were selected in our study to project future climate factors trends: (1) Beijing Climate Center Climate System Model version 1.1 (BCC CSM1.1); (2) Beijing Normal University Earth System Model (BNU-ESM) version 1; (3) the second generation Canadian Earth System Model (CanESM2), and 4) the Geophysical Fluid Dynamics Laboratory Climate Model, version 3 (GFDL-CM3). There are three Representative Concentration Pathways (RCPs) in the GCMs, i.e., RCP2.6, RCP4.5, and RCP8.5. Under the extreme RCP8.5 scenario, the area is prone to suffer more droughts and water scarcity [21]. Thus, the RCP8.5 was selected to maximize the climate change signal of the four models during the period of 2020–2099 to simulate an enhanced impact on water resources. Using such an extreme RCP scenario will be helpful for understanding the impact of vegetation regreening on water resources under extreme climatic conditions and for developing adaptive strategies and mitigating economic losses to extreme events. The four models' output variables include precipitation, near-surface air temperature, near-surface relative humidity, surface downward longwave and shortwave radiation, and near-surface wind velocity. From these variables, the future potential evapotranspiration (PET) and aridity index (AI) values in the Loess Plateau were generated. The estimated climatic data were bias-corrected and downscaled to a $0.5° \times 0.5°$ spatial resolution, and the mean value of each climatic factor in the 12 watersheds was calculated. In the models, the wind velocity is given as eastward (*ewd*) and northward (*nwd*) vector components. Therefore, the monthly wind velocity is calculated using $\sqrt{ewd^2 - nwd^2}$. GCM details are listed in Table 2.

**Table 2.** Details of the four typical CMIP5 models selected in this study.

| Models | Modeling Center/Country | Spatial Resolution/Degree | Reference |
|---|---|---|---|
| BCC CSM1.1 | Beijing Climate Center /China | 2.81 × 2.79 | Xin, Wu [22] |
| BNU ESM | Beijing Normal University, China/China | 2.81 × 2.81 | Ji, Wang [23] |
| CanESM2 | Canadian Centre for Climate Modelling and Analysis/Canada | 2.81 × 2.79 | Chylek, Li [24] |
| GFDL-CM3 | National Oceanic and Atmospheric Administration (NOAA) Geophysical Fluid Dynamics Laboratory (GFDL)/USA | 2.50 × 2.00 | Griffies, Winton [25] |

### 2.2.2. Soil Moisture and Soil Characteristic Dataset

Monthly soil moisture datasets with a spatial resolution of $0.25° × 0.25°$ for soil intervals at 0–10 cm, 10–40 cm, 40–100 cm, and 100–200 cm were obtained for the years 2000–2018 from the Noah of the National Aeronautics and Space Administration (NASA) Land Data Assimilation System (GLDAS), (https://disc.gsfc.nasa.gov/datasets). These soil datasets have demonstrated good performance in capturing soil moisture variation and have been widely used in Hydro-meteorological research [26]. In this study, soil moisture within these four layers was accumulated into the total water storage to a depth of 2 m. Moreover, water storage changes (ΔS) in the 12 watersheds were extracted by the watershed boundaries. The soil type and sand content over the Loess Plateau were obtained from the Data Center for Resources and Environmental Sciences, Chinese Academy of Sciences (RESDC) (http://www.resdc.cn). The soil type dataset was generated by the national soil field survey and classified by the traditional Genetic Soil Classification of China [27].

### 2.2.3. Land Cover Dataset

Remotely sensed land use/cover datasets (LUCDs) covering the Loess Plateau in 2000 and 2015 were obtained from the LUCDs of China [28]. The datasets with a classification accuracy above 94% were interpreted from medium resolution (30 m) Landsat series satellite images by a human-computer interactive interpretation and post-classification comparison method. Six classes, including cropland, forestland, grassland, waterbody, unused land, and build-up land, as well as 25 subclasses, were classified in the LUCDs. This study focused on forestland and grassland to investigate the evolution of vegetation in recent decades.

### 2.2.4. MODIS and Other Datasets

8-day composited MODIS $f$PAR data, covering 2000–2018, with a spatial resolution of 500 m, were obtained from the USGS Earth Resources Observation and Science (EROS) Center (https://e4ftl01.cr.usgs.gov/MOLT) to investigate the vegetation regreening. $f$PAR is defined as the fraction of photosynthetically active radiation (400–700 nm) absorbed by the green elements in vegetation canopy and has been shown to effectively characterize vegetation dynamics [29]. The 8-day data were first averaged into monthly data, and then the monthly $f$PAR in the growing season (April–October) were averaged to annual data for the 12 watersheds. A digital elevation model (DEM) with a 30 m resolution was downloaded from Geospatial Data Cloud (http://www.gscloud.cn/).

*2.3. Methods*

2.3.1. Aridity Index Calculation

AI is calculated from the ratio of *PET* to $P_{re}$ as follows:

$$AI = \frac{PET}{P_{re}} \tag{2}$$

where *PET* is the annual potential evapotranspiration (mm year$^{-1}$), and $P_{re}$ is the annual precipitation (mm year$^{-1}$). The daily *PET* is estimated by the Penman-Monteith method, which is recommended by the Food and Agriculture Organization (FAO) [30] and is expressed as follows:

$$PET = \frac{0.408\Delta(R_n - G) + \gamma \frac{900}{Ta+27} U_2(e_5 - e_a)}{\Delta + \gamma(1 + 0.34U_2)} \tag{3}$$

where $R_n$ is the daily net radiation at the canopy surface (MJ m$^{-2}$ d$^{-1}$), $G$ is the soil heat flux density (MJ m$^{-2}$ d$^{-1}$), $T_a$ is the mean daily air temperature (°C), and $U_2$ is the wind speed (ms$^{-1}$), both at a height of 2 m. $\Delta$ is the change slope with respect to the saturation vapor pressure in relation to air temperature (kPa °C$^{-1}$), and $\gamma$ is the psychrometric constant (kPa °C$^{-1}$). The variables $e_s$ and $e_a$ are the saturation and actual vapor pressure (kPa), respectively, which are calculated based on the surface relative humidity (*RH*, %) and the maximum ($T_{max}$, °C) and minimum ($T_{min}$, °C) air temperatures. $R_n$ can be estimated by the net shortwave radiation ($R_{ns}$) and net longwave radiation ($R_{nl}$).

2.3.2. Sensitivity Analysis and Contribution of Regreening to Runoff Change

The sensitivity of runoff to the landscape parameter *n* ($\varepsilon_n$) is calculated using the following equation [31,32]:

$$\varepsilon_n = \lim_{\Delta n/n \to 0}\left[\frac{\Delta R - P}{\Delta_n/n}\right] = \frac{\partial R}{\partial n} \times \frac{n}{R} \tag{4}$$

where $\partial R/\partial n$ is the partial differentiation of runoff to landscape parameter *n*. A positive value of $\varepsilon_n$ indicates an increase in runoff as *n* increases, while a negative $\varepsilon_n$ value indicates a decrease in runoff as *n* increases. For example, if $\varepsilon_n$ is equal to −0.1, then a 10% increase in *n* will cause a 1% decrease in the runoff.

The Budyko framework effectively describes the interaction of hydrology, climate, and watershed characteristics [16,18]. For a specific watershed, the water and energy balance over a long period obeys the Budyko hypothesis [33]. A variety of models were derived based on the Budyko framework, to quantify the impact of watershed characteristics on runoff. One of the most widely used models is mathematically expressed as follows [17]:

$$ET = \frac{P_{re} \times PET}{(P_{re}^n + PET^n)^{1/n}} \tag{5}$$

where *ET* is the actual evapotranspiration (mm year$^{-1}$); and *n* is an empirical parameter that represents the combined effect of climate and watershed characteristics [15]. According to previous studies, *n* is primarily determined by the underlying physical condition of the watershed, climate variation, and vegetation condition [34]. Recently, Zhang and Yang [10] reported that the ratio of $\Delta n$ to $\Delta f$PAR has a strong relationship with the aridity index; thus, the vegetation regreening, represented by *f*PAR, can be converted into $\Delta n$. The following equation reasonably calculates $\Delta n$:

$$\Delta n = 0.02 \times AI^{2.6} \times \Delta fPAR \tag{6}$$

According to Equation (4), the runoff change caused by parameter $n$ ($\Delta R_n$) can be expressed as:

$$\Delta R_n = \varepsilon_n \frac{\Delta n}{n} R \tag{7}$$

Combined Equations (6) and (7), the contribution of vegetation regreening to runoff change ($\Delta R_{fPAR}$) as follows:

$$\Delta R_{fPAR} = \Delta R_n = \varepsilon_n \frac{\Delta n}{n} R = \varepsilon_n \times \frac{\left(0.02 \times AI^{2.6}x \times^{\Delta} f_{P}AR\right)}{n} \times R \tag{8}$$

where $\varepsilon_n$ and the sensitivity of runoff to precipitation ($\varepsilon_p$) are calculated by:

$$\varepsilon_n = \frac{ln(1+AI^n) + AI^n ln(1+AI^{-n})}{n[(1+AI^n) - (1+AI^n)^y n + 1]} \tag{9}$$

$$\varepsilon_p = \frac{(1+AI^n)^{1/n+1} - AI^{n+1}}{(1+AI^n)[(1+AI^n)^{1/n} - AI]} \tag{10}$$

For a closed watershed, the water balance is expressed as:

$$ET = P_{re} - R \pm \Delta S \tag{11}$$

where $\Delta S$ is the terrestrial water storage change (mm year$^{-1}$), which describes the moisture storage changes in surface and subsurface stores. It has been established that the water storage can be regarded as constant and $\Delta S$ can safely be neglected ($\Delta S = 0$) over long time scales [35]. However, the $\Delta S$ varies significantly from 2000 to 2018 in the 12 watersheds (Figure 4). Therefore, it cannot be neglected at an annual scale, as this can lead to large uncertainties in the water balance model [26].

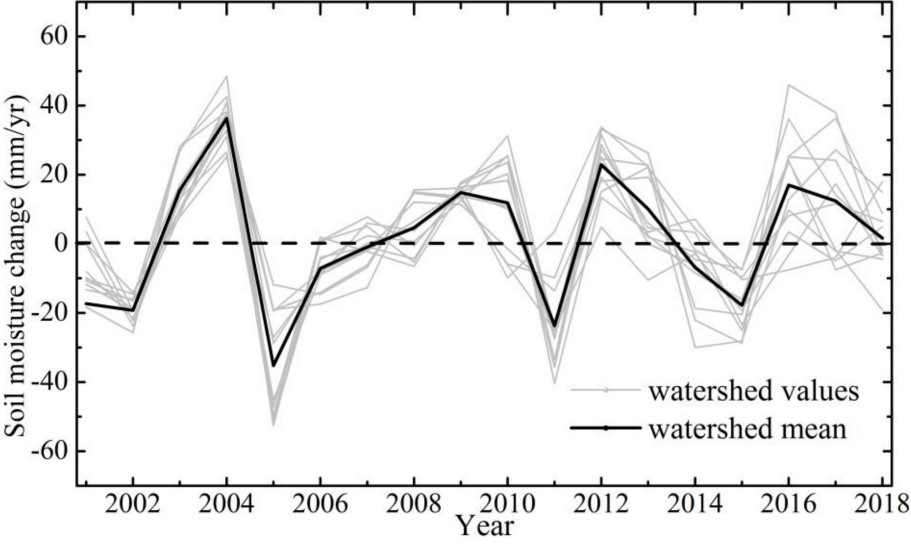

**Figure 4.** Soil moisture content changes at the depth of 2 m in the 12 middle Yellow River watersheds. The thin black lines show the 12 watershed values; the dark black line is the mean value.

For a specific period, the watershed parameter $n$ can be solved by combining Equations (5) and (11); then the contribution of vegetation change to runoff $\Delta R_{fPAR}$ can be determined. The sensitivity $\varepsilon_{fPAR}$ and the relative contribution rate $RC\_fPAR$ of regreening to runoff change can be evaluated as:

$$\varepsilon_{fPAR} = \frac{\varepsilon_n \times 0.02 \times AI^{2.6}}{n} \tag{12}$$

$$RC\_fPAR = \frac{\varepsilon_{f_{PAR}} \times \Delta fPAR}{\Delta R} \tag{13}$$

To investigate the response of the hydrological cycle to vegetation greening in the future, we designed five numerical experiments to quantify the influence of vegetation regreening on runoff variation. First, we obtained the climate condition (including PET, $P_{re}$, and AI) during 2020–2049 and 2050–2099 based on the four GCMs. Next, we presumed the vegetation would be regreening, under the future climate condition, in five scenarios. Thus, we set the change rate of *f*PAR as 0%, 20%, 40%, 60%, and 80% with respect to the mean values of 2000–2018, respectively. Based on Equation (13), the runoff change resulting from different regreening scenarios can be evaluated.

### 2.3.3. Statistical and Trend Analyses

The *f*PAR trends at the pixel scale (500 m) and the varying contribution of regreening to runoff within the 12 watersheds during 2000–2016 were identified using the Mann-Kendall (MK) test [36]. The MK test is a rank-based nonparametric method that is less sensitive to outliers than other parametric statistics and has been widely used in hydrology and climatology research [37]. The statistical Z value from the MK test was used to define whether the level of significance is reached. Critical Z values of ±1.64, ±2.58, and ±3.29 were used for the probabilities of $p$ = 0.1, 0.01, and 0.001, respectively. In this study, the MK test is applied to examine the annual trend in the AI and related climatic factors. The *f*PAR mean, trend, and Z values were mapped using the ArcGIS 10.2 software.

## 3. Results

### 3.1. Vegetation Regreening Characterized by fPAR

Figure 5 shows the spatial distribution of the mean values, trend, and the variation of *f*PAR during 2000–2018 over the Loess Plateau. Higher *f*PAR values (>0.4) are identified in the southeast of the Loess Plateau, while lower values were found mainly in the northwest (Figure 5a). The annual mean *f*PAR in the 12 watersheds follows the pattern of the Loess Plateau: larger *f*PAR watershed values in the southeast regions (Watersheds 9–13), while lower *f*PAR values were found in the northwest regions (Watersheds 1–6). The lowest averaged *f*PAR value (0.19) was found in watershed 1, while the largest *f*PAR average value, 0.26, was located in watershed 11.

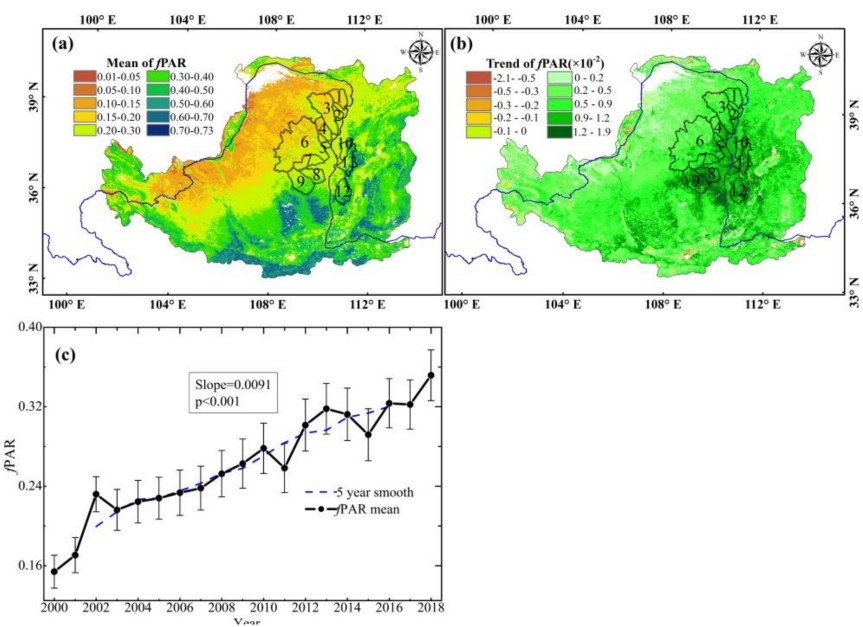

**Figure 5.** Spatial distribution of the (**a**) mean, (**b**) *f*PAR trend, and (**c**) annual variation estimated from 500 m MODIS vegetation index in the Loess Plateau from 2000 to 2018.

In over 97% of pixels over the Loess Plateau, *f*PAR showed an increasing trend (Figure 5b). Regreening trends were also observed in the central Loess Plateau, and were remarkably different from the annual mean *f*PAR pattern, particularly in the 12 selected watersheds. MK test showed that *f*PAR in all 12 watersheds increased significantly ($p < 0.001$). This was especially evident in watersheds 11, 12, 8, and 9, where *f*PAR values reached 0.0107, 0.0115, 0.0121, and 0.0125 with Z values of 5.25, 5.25, 5.12, and 5.11, respectively.

As is evident in the first three years after 2000 (Figure 5c), generally, the annual *f*PAR in the 12 watersheds experience a significant increase ($p < 0.001$) at a rate of 0.0091 year$^{-1}$. Selecting the first five years (2000–2004) as a benchmark, the change rates of *f*PAR in other years were calculated. Results for the end of 2016 showed that the *f*PAR change percentage of the 12 watersheds ranged from 43.9% to 89.1%, with a mean value of 62.6% (Figure 6). Among the 12 watersheds, 9 experienced an *f*PAR change percentage exceeding 50%, while the largest values were found in watersheds 8, 9, and 5, with values of 89.1%, 79.4%, and 76.9%, respectively.

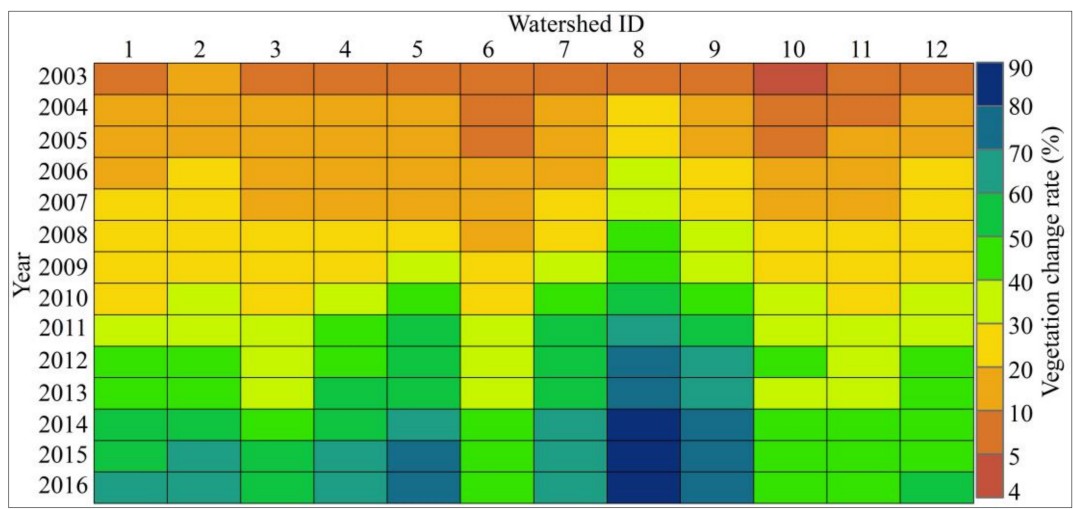

**Figure 6.** Percentage change of *f*PAR for the 12 watersheds during 2000–2018. All the change rates were compared with the base time period of 2000–2004.

*3.2. Response of Runoff to Vegetation Regreening in Recent Years*

3.2.1. Runoff Variation and Its Sensitivity to Vegetation Regreening

Figure 7 shows the variations in annual runoff, precipitation, and *f*PAR; while Figure 8 shows the runoff sensitivity to vegetation regreening throughout the 12 watersheds. During 2000–2018, the *f*PAR increased in each of the 12 watersheds, yet the runoff varied markedly from one watershed to another. Only one of the 12 watersheds, watershed 7, showed a continuous decrease in runoff as a function of vegetation regreening. In contrast, two of the watersheds, watersheds 10 and 11, demonstrated increasing runoff across the entire study period. With respect to the other watersheds, watersheds 1, 3, 4, 5, 9 and 12 exhibited an obvious decrease in runoff prior to ~2012 and then increased with continuous vegetation regreening.

The responses of runoff change to vegetation regreening throughout the 12 watersheds reveal two points: first, the response of water cycle to vegetation regreening exist strong regional difference; second, such a response also has an obvious temporal effect, i.e., exhibiting different trends at various time scales. Generally, the spatial heterogeneity in topography, slope, and elevation grow with the watershed size, which can lead to more complex effects that offset the annual runoff response to vegetation regreening. Even though the vegetation continues to regreen, the runoff cannot decrease continuously, and it can reach a tradeoff between the impact of regreening and the comprehensive hydroclimate of a specific watershed. The reduction effect in runoff by vegetation recovery has been

supported by several studies [13,38]. Our research in the middle Yellow River (Figure 7) found that reduction in runoff tended to be stable, but the runoff is prone to affect by the precipitation variation after ~10 years of vegetation regreening. To confirm the increasing impact of precipitation on runoff, we compared the Pearson coefficient (*r*) of the correlation between runoff and precipitation between 2000–2010 and 2006–2016. The results showed that the *r*-value increased significantly in all the watersheds except for watershed 7, especially for watersheds 2, 3, 5, 6, 9, 11, and 12, where *r* > 0.75. Therefore, the runoff will not sharply decline for the next several years, as it will respond exclusively to the variation in precipitation without the dramatic effect of human activity or climate change.

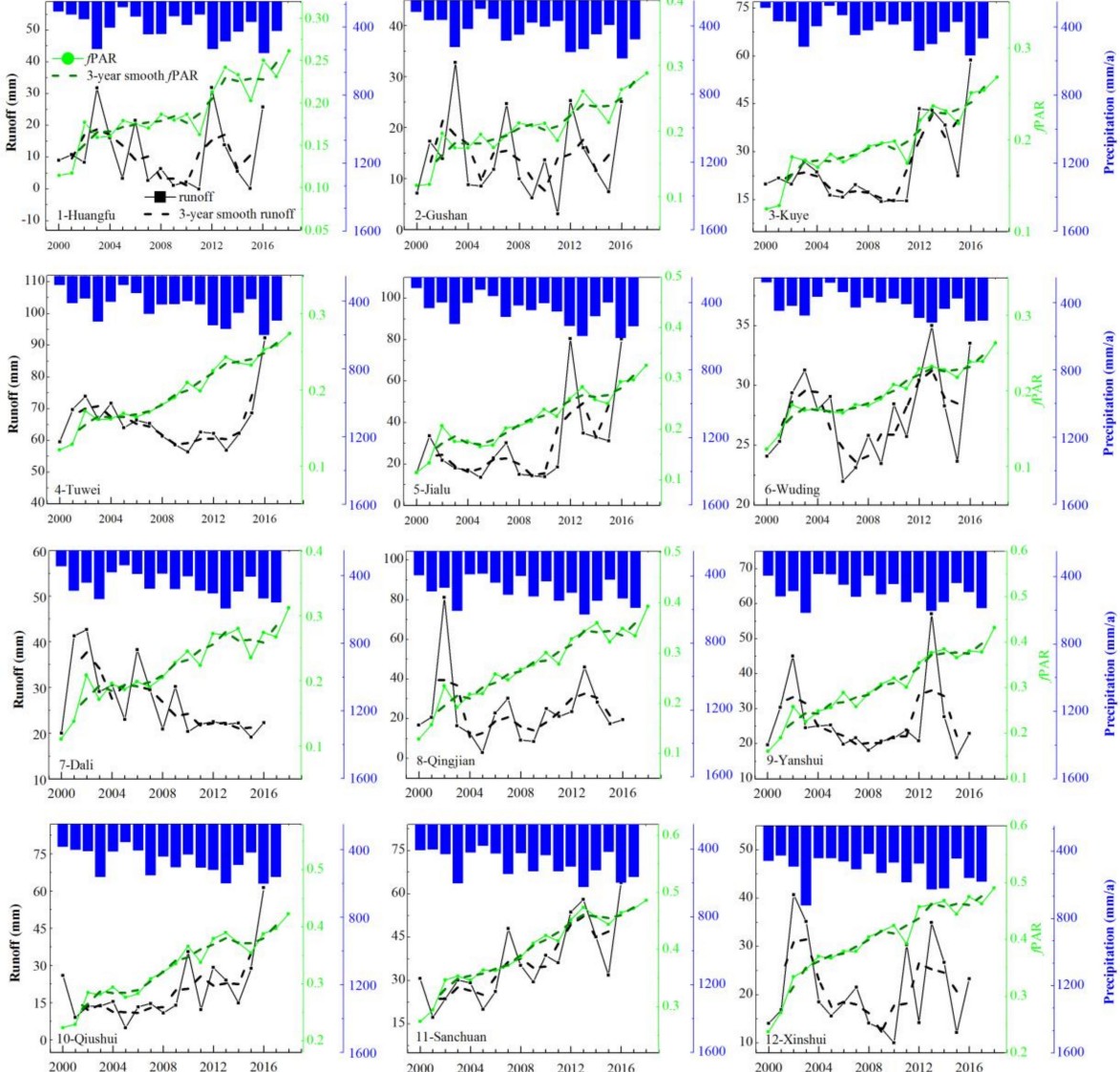

**Figure 7.** Variation in the annual runoff, *f*PAR, and precipitation throughout the 12 watersheds. The black and green lines are annual runoff and *f*PAR, respectively, and the dashed lines represent the three years' smooth average. The blue bars depict the annual precipitation.

The runoff sensitivity to regreening varies remarkably between watersheds, with values ranging from −0.325 in watershed 6 to −0.098 in watershed 12 (Figure 8a). The mean runoff sensitivity to *f*PAR value for the 12 watersheds is −0.201, which indicates that 10% of vegetation regreening can cause an average decrease in runoff of ~ 2.01%. In contrast, the runoff sensitivity to precipitation ($\varepsilon_p$) generally increases in all of the 12 watersheds, with the mean value of 3.27 (Figure 8b). This indicates that 10% increase in precipitation will lead a 23.7% increase in runoff. Comparison to the pattern of Figure 8a,b,

it is clearly can be seen that there is a complementary relationship of *f*PAR and precipitation on controlling the runoff, i.e., when the year of stronger sensitivity to *f*PAR, while weaker to precipitation, and vice versa.

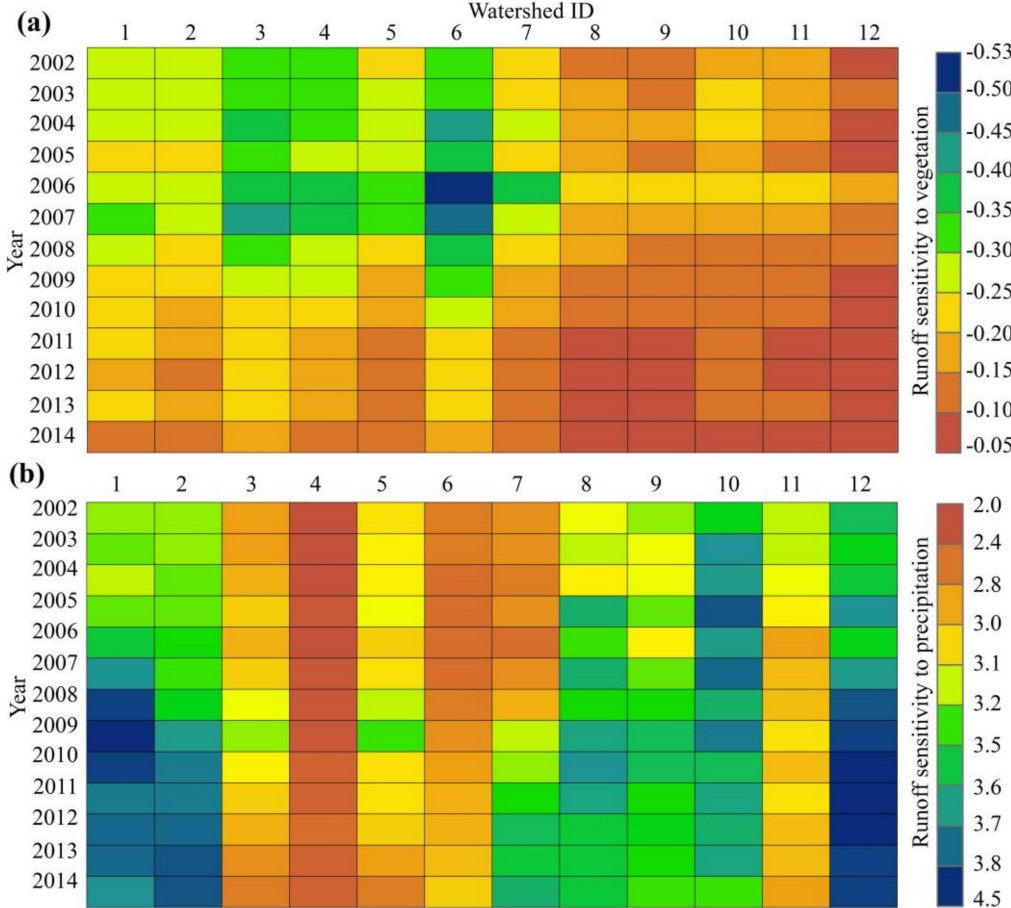

**Figure 8.** Runoff sensitivity to (**a**) regreening (*f*PAR) and (**b**) precipitation for the 12 watersheds during 2002–2014.

The absolute runoff sensitivity to regreening ($\varepsilon_{fPAR}$), for the 12 watersheds, does not exhibit a continuously increasing trend; rather, it increases until ~2006 and then decreases until 2015 (Figure 8a). The abrupt changing point of $\varepsilon_{fPAR}$ in ~2006 indicates that it reached its climax value seven years after the ecological restoration was implemented in the Loess Plateau; followed by a ten-year decline in runoff sensitivity to vegetation, which is especially evident in watersheds 8–12, with sensitivity values <−0.10. With the declining in $\varepsilon_{fPAR}$, the $\varepsilon_p$ obviously increases in most watersheds. Using the one-way variance test, we found that the difference of $\varepsilon_p$ between the period of 2002–2010 (mean of 3.16) and 2011–2014 (mean of 3.37) is significant ($p < 0.05$), which indicates that the precipitation has become the dominant factor in determining the runoff variation after around 2011 or 2012, especially in watersheds 1–9 and 12 (Figures 7 and 8b).

### 3.2.2. Impact of Vegetation Greening on Runoff

Based on the change rate of *f*PAR (Figure 6) and runoff sensitivity to regreening (Figure 8), the relative contribution of regreening to runoff change, compared with the first five years (2000–2004), was calculated and is shown in Figure 9. Generally, the contribution rate of vegetation regreening to runoff change varied among different watersheds, ranging from −6.33% in watershed 4 to −2.67% in watershed 12, with a mean rate of −5.09%.

The relative contribution rate of $f$PAR to runoff showed an increasing trend with varying amplitudes in all watersheds from 2003 to 2014 (Figure 9), which demonstrates that regreening played an increasingly important role in influencing the water resource. The most significant regreening contribution rate to runoff change appeared in watershed 5 with a value of $-0.618$ year$^{-2}$ ($p < 0.001$), while the lowest rate occurred in watershed 12, with a value of $-0.155$ year$^{-2}$ ($p > 0.05$). It is evident that the watersheds located on the west side of the Yellow River (watersheds 1–9, Figure 1), generally have larger regreening contribution rates to runoff than watersheds 10–12, which are located on the River's east side.

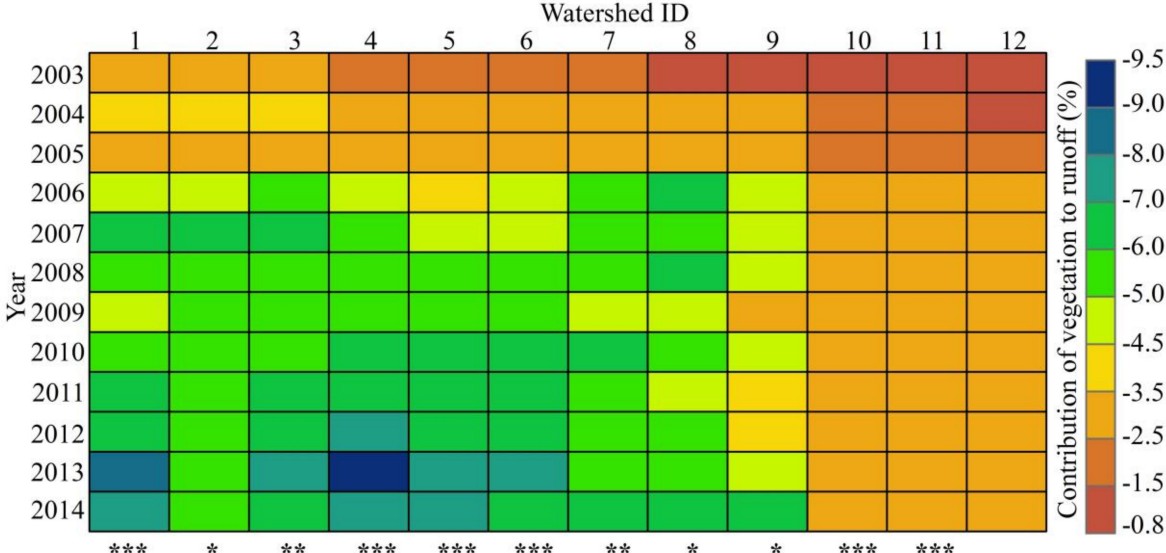

**Figure 9.** The relative contribution rate of regreening to runoff (*RC_fPAR*) in the 12 watersheds for recent years during 2003–2014. Each year's value refers to the adjacent five-year. \*, \*\*, and \*\*\* refer to the significance of the contribution rate trend at the level of $p$ = 0.1, 0.01 and 0.001, respectively.

## 3.3. Hydrological Response to Vegetation Regreening in Future Decades

Figure 10 shows the predicted variation in precipitation, *PET*, and AI throughout the 12 watersheds for the periods of 2020–2049 and 2050–2099. These trends were derived from the four GCMs (Table 2). In alignment with global climate change, precipitation exhibits an increasing trend of 3.10 mm/year ($p < 0.1$) and 1.01 mm year$^{-1}$ for previous and later periods, respectively, which means an increase in water availability throughout the 12 regreening watersheds. The *PET* also shows a significant increasing trend, with values of 2.31 ($p < 0.1$) and 1.89 ($p < 0.01$) for periods 1 and 2, respectively, which could lead to more atmospheric water evaporation, resulting in less freshwater availability. However, under the dual impact of precipitation and *PET*, the aridity index shows a slightly decreasing trend, indicating that the climate will become wetter. Such climatic conditions will be beneficial for vegetation regreening and ecological restoration.

Based on the five scenarios of vegetation regreening, the impact of regreening on runoff under the future climate projection was analyzed for the two subperiods of 2020–2049 and 2050–2099. Results show that runoff will increase by approximately 50.7% in 2020–2049 and 30.2% in 2050–2099, with no consideration of the vegetation change ($\Delta f$PAR = 0) (Figure 11). For the other four vegetation regreening scenarios, $\Delta f$PAR of 20%, 40%, 60%, and 80%, the runoff change rate will decrease from 47.2% to 37.7%, and from 26.2% to 15.3% for the two periods, which indicates that vegetation regreening will consume more water for evapotranspiration. For the sensitivity of runoff to vegetation regreening, a 20% increase in $f$PAR will cause a 3.5% and 4.1% decrease in the runoff change rate for periods one and two, respectively. Generally, runoff continues to increase for different vegetation regreening in all watersheds and scenarios in most of the watersheds for both of the subperiods.

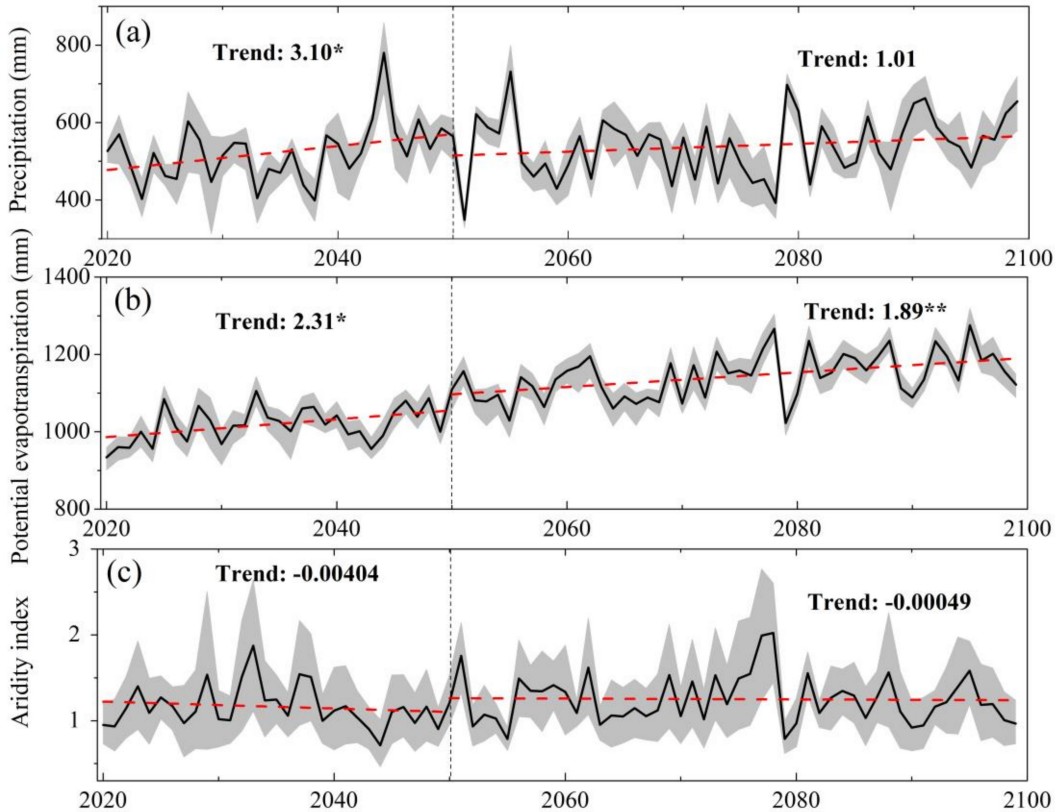

**Figure 10.** Variation and trends of (**a**) annual precipitation, (**b**) potential evapotranspiration, and (**c**) the aridity index for the 12 watersheds during the periods 2020–2049 and 2050–2099. The red dotted line refers to the trend of climatic variables, and the shaded area refers to the maximum and minimum climatic variables for the 12 watersheds.

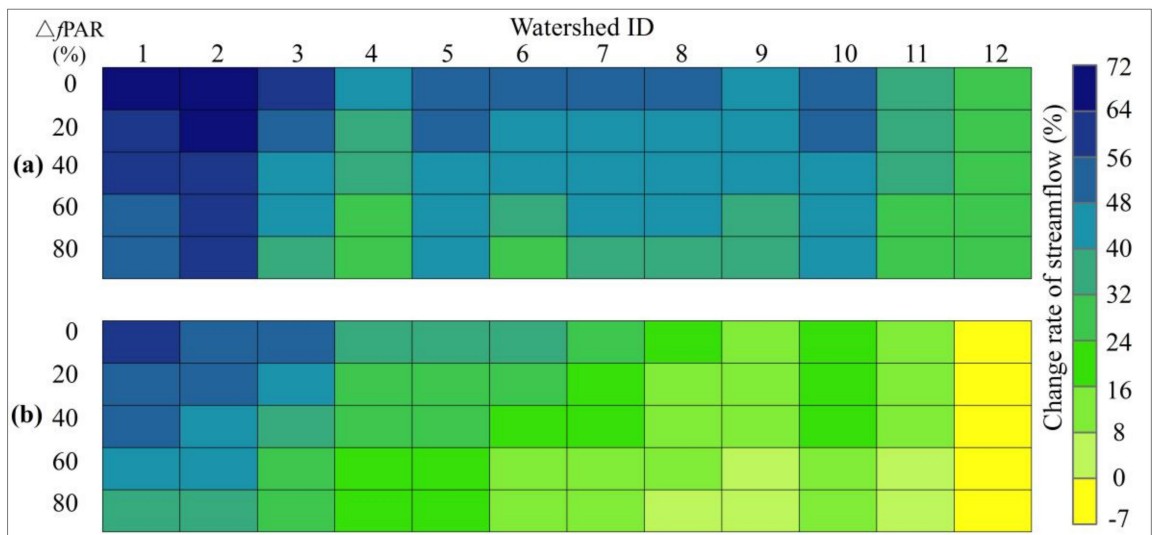

**Figure 11.** Runoff change rate for different regreening scenarios throughout the 12 watersheds for the periods (**a**) 2020–2049 and (**b**) 2050–2099 compared with the period 2000–2018.

## 4. Discussion

### 4.1. Underlying Drivers of the Rapid Vegetation Regreening after 2000 in the Loess Plateau

Previous studies have reported that large-scale vegetation changes can be easily captured by remotely sensed observations, and the remote sensing-based vegetation parameters, *f*PAR, can effectively reflect the vegetation conditions [9]. In this study, we observed the regreening trend in the Loess Plateau after the implementation of the GFGP using the MODIS data (Figures 5 and 6). Substantial increasing vegetation regreening trends were also detected throughout the 12 watersheds since 2000 (Figures 5 and 6), which are highly consistent with the time and key regions of the GFGP implementation. To further explore the underlying vegetation regreening factors, the change rates of forest and grass over the Loess Plateau and 12 watersheds between 2000 and 2015 were compared (Figure 12 and Table 3).

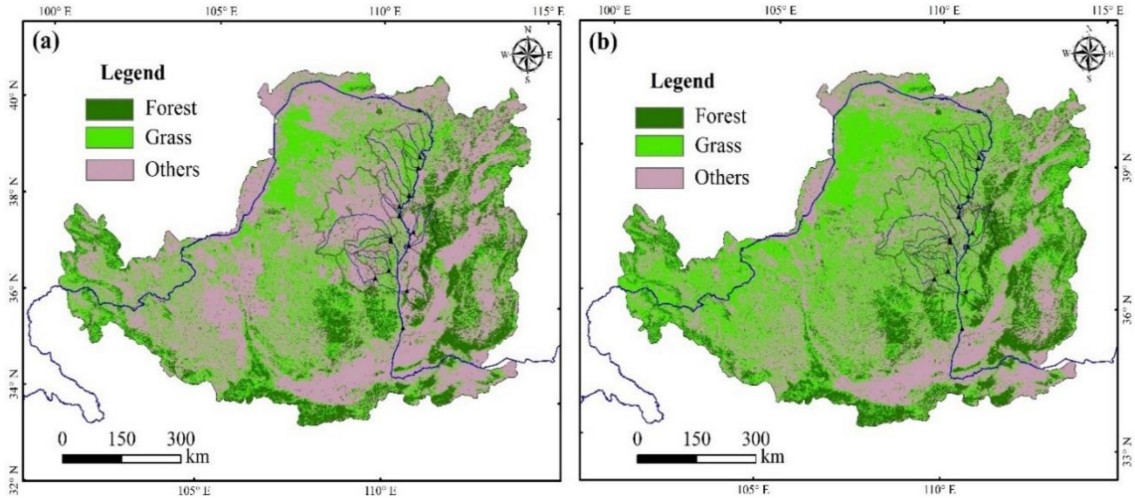

**Figure 12.** Spatial distribution of vegetation cover in (**a**) 2000 and (**b**) 2015 in the Loess Plateau.

With respect to forest, 10 of the 12 watersheds exhibited a positive change rate. The largest forest change rate occurred in watershed 8, with a value of 55.7%, followed by those of watersheds 5, 6, 9 and 3, with values of 39.1%, 29.3%, 29.3%, and 28.0%, respectively. Overall, the mean change rate of the 12 watersheds reached 12.7%; much higher than other locations in China and around the world [39]. For grass, all the watersheds, except for watershed 2, showed a positive rate from 2000 to 2015 and depicted a larger average change rate (123.1%) as compared to the forest (Table 3). The two largest change rates appear in watersheds 5 and 12 with values of 828.6% and 795.4%, respectively. The next largest change rates were watersheds 11, 4, 6, 9 and 10, with values of 239.6%, 175.3%, 169.5%, 157.1%, 147.4%, respectively. Therefore, the large change rates of the forest and the grass throughout the 12 watersheds were a result of GFGP implementation. After 2000, GFGP implementation became the dominant driver for the rapid regreening of key areas in the Loess Plateau.

**Table 3.** Vegetation (Forest and Grass) change rates throughout the 12 watersheds in 2000 and 2015.

| Types | Year | 1 | 2 | 3 | 4 | 5 | 6 | 7 | 8 | 9 | 10 | 11 | 12 |
|-------|------|---|---|---|---|---|---|---|---|---|----|----|----|
| **Forest** | 2000 | 137 | 73 | 357 | 109 | 23 | 702 | 278 | 431 | 546 | 560 | 1854 | 1153 |
| | 2015 | 164 | 76 | 457 | 111 | 32 | 908 | 329 | 671 | 706 | 559 | 1848 | 1154 |
| | change (%) | 19.7 | 4.1 | 28.0 | 1.8 | 39.1 | 29.3 | 18.3 | 55.7 | 29.3 | −0.2 | −0.3 | 0.1 |
| **Grass** | 2000 | 1287 | 646 | 3423 | 530 | 56 | 3729 | 839 | 745 | 1009 | 308 | 288 | 218 |
| | 2015 | 2083 | 637 | 5339 | 1459 | 520 | 10048 | 1434 | 1363 | 2594 | 762 | 978 | 1952 |
| | change (%) | 61.8 | −1.4 | 56.0 | 175.3 | 828.6 | 169.5 | 70.9 | 83.0 | 157.1 | 147.4 | 239.6 | 795.4 |

### 4.2. Implications of the Impact of Vegetation Regreening on Eco-Hydrology in Water-Limited Regions

It has been established that reforestation can decrease runoff and deforestation can increase runoff at different magnitudes [13]. Furthermore, our findings demonstrated that the runoff sensitivity to regreening was large after the first ten years of reforestation (2000–2009), particularly in watersheds 1–7 (Figure 8). However, after 2010, such sensitivity tends to significantly differ among the 12 watersheds. As shown in previous studies [10,18], the underlying cause of runoff sensitivity variations to vegetation can be attributed to the combined effect of climate, vegetation, anthropogenic interference, soil characteristics, or topography. Since the soil and topography do not change frequently over the last two decades, thus sensitivity is mainly affected by climate, vegetation, and human activities. Climate change, particularly in precipitation, can significantly impact the runoff variation. We find that the impact of vegetation on runoff will decrease with increases in precipitation and vice versa in the 12 watersheds (Figures 7 and 8), and this is consistent with previous studies conducted in different climate regions [10,40]. It is noteworthy that larger sensitivity of runoff to vegetation in arid conditions, such as years or seasons with high temperatures and low precipitation, will accelerate water scarcity. This can cause great damage to agricultural production and societal economy if the vegetation continuously regreens in water-limited areas [21].

The relatively small negative contribution rate of $f$PAR change to runoff change indicates that although the vegetation regreening reduces runoff, it was not the dominant factor for runoff variation in the Loess Plateau in the past two decades (Figure 9). The contribution values of vegetation to runoff in our study are different from those of previous studies [41,42] for the following reasons: Firstly, there was an increase in precipitation in recent years. Compared with the period of 1980–1999, annual precipitation increased by nearly 9% during 2000–2017, and even reached 21% during 2010–2017. Secondly, previous work that did not give an explicit equation for the vegetation change and runoff variation certainly assumed that the change in runoff attributed to the change of the parameters of $n$ [43] or $w$ [41] in the Budyko framework. However, this assumption is inappropriate due to the number of factors influencing those parameters [44]. For a longer temporal scale, the runoff will not be stable in the coming 30 years in all scenarios, in response to the gradually increasing precipitation trend and larger increasing $PET$ rate (Figure 10).

The large changes in $f$PAR for the 12 watersheds demonstrate that the vegetation and degraded ecosystem have been significantly recovered since the implementation of ecological restoration, and severe soil losses have rapidly decreased. However, the central government of China recently made a new plan to invest another 9.5 billion USD for the purpose of intensifying the GFGP in the Loess Plateau by the middle of the 21st century [1]. This new strategy has been the center of ample attention and debate. A primary concern is whether an infinite investment in regreening will enhance negative eco-hydrological effects, such as a decline in soil moisture [3] and an increase in drought frequency [45].

The GFGP is a trade-off between vegetation regreening and eco-hydrological balance [6]. On one hand, reforestation or re-vegetation plays a positive role in reducing sandstorms, soil erosion and floods, and improving vegetation carbon sequestration and food supply [46]. On the other hand, the vegetation regreening poses a threat to regional freshwater availability [45] and facilitates drought [3] in water-limited regions such as the Loess Plateau. For a long period, areas around the middle Yellow River, especially the 12 selected watersheds in our study, have been the most sediment-laden regions in the world, and have accounted for approximately 90% of sediment loss in the Loess Plateau [4]. This high load of sediment from the Loess Plateau deposited in the lower reaches of the Yellow River (Figure 1), formed a riverbed at a height of 20 m above the surrounding land surface. Fortunately, the rapid regreening of vegetation in recent years over the Loess Plateau has significantly reduced the soil erosion, and the sediment discharge has been successfully reduced to the historical level of one thousand years ago [6]. Therefore, the current regreening program is beneficial for regional ecological balance and human–nature sustainable development; however, continuous vegetation regreening and its impact on eco-hydrology still requires future research.

*4.3. Uncertainties and Prospects*

In this study, the influence of regreening on runoff has been identified in terms of the most afforested 12 watersheds over the Loess Plateau. However, this work also had to contend with some inherent limitations. First, the selection of the CMIP5 GCMs simulations to project future climatic conditions automatically introduces uncertainties. While the GCMs are the only known effective tool to depict the complex processes and to determine future climate change, they also exhibit various sources of uncertainty, e.g., coarse spatial resolution, insufficient description of the subgrid-scale processes, and model structures [47]. Earth's orbit, precession, and tilt may be periodically perturbed by the natural solar/astronomical oscillations of solar system [48]. Perturbations of this type can cause significant solar radiation redistribution and climate change. However, the mechanisms related to this planetary oscillation are absent from the CMIP5 GCMs, which calls into question whether they can be relied on to accurately interpret and predict future climate change. In our study, four typical GCMs under the RCP8.5 scenario were employed to project an extreme climate scenario. Although we averaged their projected climatic values for the two long periods (2020–2049 and 2050–2099) to reduce errors from different models, other errors may still exist. However, we mainly focused on the general trend of climate change (Figure 10) and its impact on the relative change of runoff in our study and the usage of the GCMs characterizing the direction of climate change is valuable. Second, according to the Budyko hypothesis, several models were derived from the original form [14,16–18]. The single one (Choudhury-Yang model) was used in our study, which may introduce uncertainties in the follow-up analysis. However, several studies have reported that all of the equations share a similar curve pattern with no significant differences [31]. Such uncertainties caused by equation selection should be limited. In addition, the runoff can be affected by several factors, including vegetation, climate (precipitation and PET) and human interference. In this study, we focused mainly on the impact of vegetation regreening on runoff variation with the assumption that vegetation is independent of other factors. This assumption may introduce some uncertainties as those factors can interact with each other.

Regardless of the uncertainties mentioned above, our preliminary work presents a valuable case for quantifying the vegetation regreening change and its impact on runoff.

Our intention in this paper is to use these findings as a baseline for follow-up research to investigate the interaction and feedback mechanisms between the vegetation regreening and hydrological processes. A distributed hydrological model [49], coupled water and energy balance and vegetation change, will be introduced for paired watersheds experiment. This will help for understanding the relative independent contribution of hydro-meteorological variables, such as precipitation, PET, canopy interception, vegetation transpiration, soil evaporation, snow depth, and groundwater, etc., to runoff change.

## 5. Conclusions

Using the *f*PAR, water-energy balance model, and GCMs' climate change, we quantitatively assessed the responses of the hydrological cycle to vegetation regreening throughout 12 watersheds of the Loess Plateau in both the past and future. Results showed that: (1) the *f*PAR throughout in the 12 watersheds showed a considerable regreening trend, and the rapid increase in forest and grass coverage was the dominant driver for the regreening; (2) the trend of runoff sensitivity to regreening underwent an abrupt reversal from increasing to decreasing in 2006 and has declined ever since. The increasing trend in *RC_f*PAR was significant in 11 out of 12 watersheds ($p < 0.05$), which demonstrates that regreening is playing an increasing role in controlling the runoff variation; and (3) climate change will increase the runoff by only 50.7% and 30.2% during the periods of 2020–2049 and 2050–2099, respectively, based on climate projections. However, runoff is reduced in different regreening scenarios, with a mean decrease value of 3.5% and 4.1% per 20% increase in *f*PAR for the periods of 2020–2049 and 2050–2099, respectively.

This study presents a very promising case for using the moderate spatial resolution and temporal frequency satellite-based vegetation index to evaluate the influence of vegetation change on the water cycle. In the future, we will continue to investigate the impact of vegetation change on soil moisture

and evapotranspiration, including its three components (canopy interception, leaf transpiration, and soil evaporation), by using satellite-based vegetation index of leaf area index (LAI) and distributed hydrological models, and finally to reveal the physical mechanisms for the eco-hydrological effect of vegetation change.

**Author Contributions:** Y.L. performed the data processing as well as wrote the manuscript. D.M. provided the ideal and constructive suggestions towards the whole project. A.F. helped to download and process the *f*PAR datasets. T.S. polished our manuscript with professional English and provided useful comments on the results.

**Funding:** This research was jointly funded by the Natural Science Foundation of China grant number [41701019], the funding from Youth Innovation Promotion Association of Chinese Academy of Sciences (2017277), National Key Research and Development Program of China grant number [2018YFC1509002], the Startup Foundation for Introducing Talent of Nanjing University of Information Science and Technology (NUIST) grant number [2017r069], and the China Scholarship Council grant number [201809040009].

**Acknowledgments:** The authors give many thanks to the Hydrological Bureau of the Yellow River for providing the runoff data of the 12 watersheds. We appreciate the writing service provided by the Miller Writing Center of Auburn University (http://wp.auburn.edu/writing/writing-center/) to fix mistakes of grammar and selection of words, and by TopEdit LLC (www.topeditsci.com) for the linguistic editing and proofreading during the preparation of this manuscript. We also appreciate assistant editor Brenda Zhao and the two anonymous reviewers for their efforts and constructive comments on our manuscript.

**Conflicts of Interest:** The authors declare no conflict of interest.

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
