# Peer review of "Will Human-Induced Vegetation Regreening Continually Decrease Runoff in the Loess Plateau of China?"

_forests, doi:10.3390/f10100906_

Round 1
Reviewer 1 Report
Please respond to the comments:
L 117 according to which classification the division of catchment soils should be given;
L 171 GCM, provide full name, abbreviations in the text should be described;
L173 The selection of the most extreme, pessimistic RCP8.5 scenario should be discussed (for example in section 4.3);
In addition, projections for 2020-2049 and 2050-2099 may be subject to uncertainty (L.341).
Mainly because of other unpredictable factors in the future. For example: changes in the shape of the Earth's orbit, precession and nutation (changes in the orientation of the axis and its tilt to the plane of the orbit). According to literature, this is a change in energy supply by about 0.2% / year.
Author Response
Dear professor,
Thank you very much for giving us the constructive suggestions and comments to our manuscript titled with “Will human-induced vegetation regreening continually decrease runoff in the Loess Plateau of China?” (ID: forests-583889). According to your comments, we have revised the manuscript carefully and marked the modification in RED font and new texts so that the changes are readily visible.
A detailed point-by-point response documenting all the changes is attached with this letter. The co-author named Dr. Tayler Schillerberg and several native speakers from the writing service of Auburn University edited the revised manuscript. In addition, we called for language editing service (https://www.topeditsci.com/home) to polish our language. We hope that the revised manuscript will meet the publication standards of Forests.
Thank you again, and we are looking forward to hearing from you.
Sincerely yours,
Yanzhong Li
On behalf of all authors

Reviewer 2 Report
I removed some of the unnecessary text and placed various comments in the text of the manuscript to improve the English.
Figure 7 shows some interesting results that were not correctly interpreted in my mind. All 12 watershed revealed greening trends. This is good.
However, the correlation to decreased discharge is open to speculation and reinterpretation. Only one of the 12 watersheds, watershed 7, shows a continued decrease in stream discharge. A number of the watersheds reveal decreased flow to 2010, watersheds 1, 3, 4, 9, but not as many as the authors claim. Discharge in these watershed then increase to the present day, opposite to their stated inverse correlation between greening and discharge. Two of the watersheds reveal increasing stream discharge, watershed 10 and 11, through the entire study.
Finally, many of the discharge records are noisy, jump up and down from one year to the next. Is this variability due to parallel changes in precipitation or something else? The authors have the precipitation data but they do not investigate this possibility. For example, almost every record reveals higher discharge during 2011 and/or 2012. They need to check this, as it might substantiate their claim but providing a reason why a few years do NOT follow the general decreasing trend. It does not parallel changes in the greening of the landscapes as this followed a uniformly increasing trend in all watersheds. Because the main focus of this paper, the inverse correlation between forestation and discharge, is questionable in my mind based on what I see in Figure 7, I cannot provide a positive review for this paper.
I openly speculate on the validity of the sensitivity test. Perhaps it magically removed rainfall variability from the raw data? If yes, then this was not detailed in the manuscript.
I stopped my careful review of the manuscript at the end of section 3.2.1.
The manuscript could use another editor to fix numerous problems in the grammar and selection of words. I added sticky notes next to the worse sections and sentences, and deleted some prose (or entire paragraphs, that I thought did not add to the manuscript.
It looks like the authors did NOT complete the Funding section of the manuscript – line 491.

Author Response
Dear reviewer,
Please accept our most sincere thanks for giving us the constructive suggestions and comments to our manuscript titled with “Will human-induced vegetation regreening continually decrease runoff in the Loess Plateau of China? ” (ID: forests-583889). According to the comments, we gain lots of valuable experience in research work. In the revised manuscript, we have marked the modification in RED font and new texts so that the changes are readily visible.
A detailed point-by-point response documenting all the changes is attached with this letter. The co-author named Dr. Tayler Schillerberg and several native speakers from the writing service of Auburn University edited the revised manuscript (http://wp.auburn.edu/writing/writing-center/). In addition, we called for language editing service (https://www.topeditsci.com/home) to polish our language. We hope that the revised manuscript will meet the publication standards of Forests.
Thank you again, and we are looking forward to hearing from you.
Sincerely yours,
Yanzhong Li
On behalf of all authors

Round 2
Reviewer 2 Report
I found some typos and other errors with grammar:
Suggested solutions
Line 31: change after while with afterwards
Line 32: delete be stable and
Line 67: insert the between in hydrology
Line 69: insert infiltration after leaf evaporation
Line 80: delete words from relationship to does
Line 86: delete as study
Line 87: area
Line 112: place 12 before selected for consistency with rest of manuscript
Line 153: replace Futhermore with From these variables
Line 257: replace northeast with northwest
Line 270-271: the sentence is unclear, rewrite entire sentence
Figures 5, 8, 9, 11: Be consistent, Watershed ID or Basin ID but do not flip flop from one to the next.
Figure 11: Watershed not Waershed.
Major issue still present in manuscript:
In the section 3.2.1, the authors try to address my primary concern about the impact of changes in precipitation on runoff. Adding the precipitation data to Figure 7 helps. Looking at this Figure, I believe that it clearly shows a connection between runoff and precipitation. More precipitation, more runoff, especially after 2012. The authors provide some r, correlation values, for runoff to precipitation, proving my criticism. However, it appears that runoff is more sensitive to changes in precipitation than re-greening, as re-greening steadily increases in all watersheds, but runoff does not.
The manuscript would therefore be significantly improved if the authors could take one more step and attempt to remove the variability in runoff due to changes in precipitation. The residual should then be evaluated if it follows their conclusions, less runoff with more re-greening.
Author Response
Dear reviewer,
Thank you so much for giving us further guidance on our paper. According to your good suggestion about some spell and grammar mistakes, we have revised them one by one. In addition, the authors also checked the manuscript carefully several times to make sure no such mistakes, please see our revised manuscript in RED font for details.
To quantitatively analyze the runoff sensitivity to precipitation, we redrew Figure 8 to show the changing pattern of runoff sensitivity to precipitation and fPAR. You can see the change in our revised manuscript and response to comments.
Please accept our sincere gratitude for the efforts you have made for our paper. We are looking forward to hearing from your early favorable reply.
Sincerely yours,
Yanzhong Li
On behalf of all authors
